# An Integrated Analysis Approach to Unravel the *Aspergillus* Community in the Hospital Environment

**DOI:** 10.3390/jof11090626

**Published:** 2025-08-26

**Authors:** Laura García-Gutiérrez, Emilia Mellado, Pedro M. Martin-Sanchez

**Affiliations:** 1Grupo Microbiología Ambiental y Patrimonio Cultural, Instituto de Recursos Naturales y Agrobiología de Sevilla (IRNAS), Consejo Superior de Investigaciones Científicas (CSIC), Avda. Reina Mercedes 10, 41012 Seville, Spain; lauragg@irnas.csic.es; 2Laboratorio de Referencia e Investigación en Micología, Centro Nacional de Microbiología, Instituto de Salud Carlos III (ISCIII), Ctra. de Pozuelo 28, Majadahonda, 28220 Madrid, Spain; emellado@isciii.es; 3Centro de Investigación Biomédica en Red—Enfermedades Infecciosas (CIBERINFEC-CB21/13/00105), Instituto de Salud Carlos III (ISCIII), Majadahonda, 28220 Madrid, Spain

**Keywords:** indoor fungi, nosocomial infections, mycobiome, healthcare settings, air, surfaces, cultivation, DNA metabarcoding, quantitative PCR

## Abstract

The genus *Aspergillus*, widely distributed across natural and urban environments, may cause allergies and opportunistic infections such as chronic or invasive pulmonary aspergillosis. Its high pathogenic potential for immunocompromised patients, together with the alarming increase of azole resistance reported in clinical and environmental isolates, claims urgent actions to assess and control the *Aspergillus* community in hospital environments. To contribute to that, here, we combine a large environmental survey covering numerous air and surface samples from different zones of three hospitals in Spain, with an integrated approach including general and selective culture- and eDNA-based analyses. Despite the high prevalence of *Aspergillus* observed, present in almost all indoor zones (mostly in air but also on surfaces) of the three hospitals, its relative abundance in the whole fungal community was limited and dependent on the used methods, with median values ranging from 1.4% (eDNA data) and 6.8% (cultivation at 28 °C) to 28.3% (cultivation at 37 °C). Remarkably, the most protected zones (intensive care units) showed the highest proportion of *Aspergillus* eDNA sequences. A total of 32 species belonging to 10 *Aspergillus* sections were molecularly identified, including well-known causal agents of invasive pulmonary infections such as *A. fumigatus*, *A. flavus*, *A. terreus*, *A. niger*, *A. oryzae*, *A. sydowii*, and *A. tubingensis*. This highlights the importance of such environmental assessments for monitoring and controlling the fungal burden in hospitals.

## 1. Introduction

*Aspergillus* species are ubiquitous saprobic fungi able to grow under a wide range of environmental conditions (temperature, pH, water activity, and nutrient availability), which allow them to colonize diverse niches such as soil, plants, decaying vegetation, and air, since plant debris and compost are the main reservoirs of this genus in the environment [1,2].

Some *Aspergillus* species have been described as allergic agents and/or opportunistic human pathogens causing invasive pulmonary aspergillosis (IPA) and chronic pulmonary aspergillosis (CPA). Although *Aspergillus fumigatus* is the most common causal agent of these diseases, other *Aspergillus* species, such as *A. flavus*, *A. terreus*, *A. niger*, and *A. nidulans*, can also cause IPA in susceptible patients [1]. *A. fumigatus* conidia spread very efficiently by the air due to their hydrophobicity and melanin in the cell wall, which protects them from the UV radiation [2]. These conidia have diameters ranging from 2 to 5 μm and, therefore, can be easily inhaled and reach the alveoli in the lungs. Under immunocompromised conditions, they might grow, infecting lung tissues and developing IPA [3]. It is estimated that 2,113,000 people suffer from IPA each year around the world, with a mortality rate of 85.2% [4]. Moreover, during the last viral pandemics, both COVID-19-associated pulmonary aspergillosis (CAPA) and influenza-associated pulmonary aspergillosis (IAPA) cases have increased considerably, affecting around 10–20% of patients with severe viral infections in intensive care units (ICUs), with a mean mortality rate of 50% [5]. Triazole drugs are the antifungal of choice for prophylaxis and first-line treatment of *Aspergillus* infections [6]. However, azole-resistance reports in both clinical and environmental isolates of *A. fumigatus* have continuously increased in the last decades [7]. The omnipresent use of azoles in agriculture, horticulture, and the forest industry is thought to be the main cause of the emergence of azole cross-resistant *A. fumigatus* isolates from the environment [8,9]. This critical situation has motivated the inclusion of this species in the critical priority group of the first fungal priority pathogen list developed by the World Health Organization (WHO) in 2022 [10].

Considering the abundance of immunocompromised patients in hospitals, especially in ICUs, it is necessary to carry out environmental microbiological monitoring protocols that control the fungal bioburden with special attention to relevant opportunistic pathogens such as *Aspergillus* and *Candida*. Routine environmental assessments in hospitals are mainly based on culture-dependent analyses of air and surface samples, counting of colony-forming units (CFU) grown on culture media, and referring them to the sampled units (e.g., cubic meter of air or square centimetre of surface) [11,12]. However, these methods only reflect the microorganisms that are in a culturable state, underestimating the total number of the microorganisms present in the hospital environment [13]. Environmental DNA (eDNA)-based techniques, such as real-time quantitative polymerase chain reaction (qPCR), high-throughput amplicon sequencing (DNA metabarcoding), or shotgun metagenomics, can be powerful, effective, and sensitive alternatives for detection, quantification, and characterization of the fungal burden in hospitals, though they cannot discriminate viable from non-viable microorganisms [14,15].

However, eDNA-based assessments in healthcare settings are still scarce [16] and should be evaluated further. To fill this gap, the *Mycospitalomics* project has recently studied the environmental fungal communities in three hospitals in Spain, combining a large survey, including replicated air and surface samples from four different zones collected in two contrasting seasons, with environmental monitoring, particle counting, culture-dependent, and eDNA (DNA metabarcoding and qPCR) analyses [17,18]. In this study, we use the valuable fungal datasets from this project to address two specific objectives: (i) to improve the knowledge about the environmental *Aspergillus* community in Spanish healthcare settings through an integrated methodological approach, as well as (ii) to evaluate the clinical relevance of the *Aspergillus* species identified.

## 2. Materials and Methods

### 2.1. Sampling

Since the detailed sampling scheme has been previously published [17], we provide a condensed version here. Two air sampling campaigns were performed in three tertiary-care hospitals from Spain (Virgen del Rocío in Seville—“VR”, La Fe in Valencia—“LF”, and Severo Ochoa in Madrid—“SO”) in winter and autumn 2023. Four hospital zones were compared in each hospital: (“OUT”) pedestrian paths outside the buildings, (“IA”) unprotected admission halls and waiting rooms, (“IB”) clean regular patient rooms, and (“IC”) ICUs with high-efficiency particulate air (HEPA) filters. A total of 72 air samples were collected at 70–150 cm height using appropriate equipment for each approach: the volumetric SAS Super Duo 360 impactor (VWR International, Milan, Italy) for cultivation and the cyclonic Coriolis µ sampler (Bertin Technologies, Montigny-le-Bretonneux, France) for eDNA analyses. A total of 126 surface samples were collected: 54 surface samples from the intake ventilation grids (vent), 54 high-touch surface samples (HTS), i.e., combined samples from bedrails, mattress zippers, bed tables, drip stands, and monitors at IB and IC zones, or counters, seats, touch screens, and vending machines at IA zones, and 18 settled dust samples from outdoor surfaces, such as window sills and railings. All surface samples (“vent”, “HTS”, and “dust”) were collected using sterile 3M sponge-sticks soaked with neutralizing buffer (3M, St. Paul, MN, USA).

### 2.2. Cultivation

In winter, airborne spores were cultivated on two general fungal media, Sabouraud dextrose chloramphenicol agar (SDCA; Labkem, Barcelona, Spain) and dichloran rose bengal chloramphenicol agar (DRBC; Scharlau S.L., Sentemenat, Spain), and incubated at 28 °C in darkness. In autumn, to selectively cultivate potentially pathogenic *Aspergillus* species, air samples were plated on SDCA with and without 4 mg/L voriconazole and incubated at 37 °C. Surface (sponges) samples were immediately processed in biosafety cabinets after reception at the laboratories, adding 20 mL PBS buffer with 0.05% Tween 20 to the bags, followed by 2 min of hand massage and orbital shaking at 180 rpm for 10 min. Then, 100 μL of the suspension were cultivated using the culture media and incubation conditions detailed above. After 1-week incubation, those colonies showing typical morphological features of *Aspergillus* were isolated in pure culture on SDA and selected for molecular identification and preservation in culture collection (biomass in 30% glycerol at −80 °C) [17].

### 2.3. Molecular Identification of Isolates

The identification of *Aspergillus* isolates was based on PCR amplification and Sanger sequencing of two markers, the ribosomal internal transcribed spacer (ITS) regions using the primers ITS1 5′-TCCGTAGGTGAACCTGCGG-3′ and ITS4 5′-TCCTCCGCTTATTGATATGC-3′ [19], and a portion of the β-tubulin gene using the primers Bt2a 5′-GGTAACCAAATCGGTGCTGCTTTC-3′ and Bt2b 5′-ACCCTCAGTGTAGTGACCCTTGGC-3′ [20]. Protocols for genomic DNA extraction and PCR were thoroughly described by Garcia-Gutierrez et al. [17]. Positive PCR products were sent to Macrogen Spain (Madrid) for purification and Sanger sequencing by duplicate using the primers ITS1 and ITS4 for ITS, and Bt1_seq 5′-AATTGGTGCCGCTTTCTGG-3′ and Bt4_seq 5′-AGCGTCCATGGTACCAGG-3′ for β-tubulin [21]. Quality checks and alignment of the sequences were performed using the BioEdit v 7.7.1 software [22]. Consensus sequences were deposited in the GenBank database from the National Center for Biotechnology Information (NCBI). Molecular identification was assigned using the basic local alignment search tool nucleotide (BLASTn) from NCBI and further curated using Mycobank database (https://www.mycobank.org/; accessed on 13 May 2025).

### 2.4. DNA Extraction from Air and Surface Samples

Air samples collected in Coriolis cones (15 mL PBS buffer with 0.005% Triton X) were filtrated using sterile 25 mm cellulose acetate filters with 0.2 µm pore diameter (Sartorius Stedim biotech GmbH, Göttingen, Germany) in a biosafety cabinet class II. Surface sponges were processed as follows: adding 20 mL PBS buffer with 0.05% Tween 20 to the bags, 2 min of hand massage, orbital shaking at 180 rpm for 10 min, and centrifugation at 3500 rpm for 15 min. Air filters and pelleted cells from surfaces were transferred to the PowerBeed Pro tubes from the DNeasy PowerSoil Pro Kit (Qiagen GmbH, Hilden, Germany), and their DNA was extracted following the manufacturer’s instructions

### 2.5. DNA Metabarcoding

The fungal libraries, from both air and surface samples, were constructed employing a two-step PCR protocol, as previously reported [18], where all details (primer sequences, PCR reagents, cycling parameters, etc.) are shown. Briefly, for the first PCR, eDNA extracts were added to duplicate reactions using primers that include Illumina sequence adaptors attached to fungi-specific sequences (forward ITS3-Mix2 and reverse ITS4-cwmix1-2). The resulting amplicons cover part of the 5.8S rRNA gene and the entire internal transcribed spacer 2 (ITS2). Positive reactions were subsequently pooled and purified using the Agencourt AMPure XP beads (Beckman Coulter Inc., Brea, CA, USA) following a standard protocol [23]. Purified PCR products from the first PCR served as template for the second PCR, which introduced sample-specific primer combinations that include 8 bp barcodes (i5 or i7 index) and complete Thruplex adapters for Illumina sequencing. Similarly, second PCR products were cleaned with AMPure XP beads and subsequently quantified using a Qubit dsDNA HS assay (Invitrogen—Life Technologies, Eugene, OR, USA) following the manufacturer’s instructions. PCR-positive samples were pooled equimolarly together technical replicates, negative samples (unused air filters and sponges, as well as blanks at DNA extraction and PCR steps), and a mock community sample (equimolar DNA mixture from four fungal isolates) and subsequently sent to the Genomics Unit at IPBLN-CSIC (Granada, Spain) for Illumina MiSeq PE300 v3 sequencing. The resulting raw sequencing data are available at the European Nucleotide Archive (ENA), EMBL-EBI, under accession no. PRJEB86993 (https://www.ebi.ac.uk/ena/browser/view/PRJEB86993; accessed on 2 July 2025).

All details about the bioinformatics pipeline are available in [18] and its associated Zenodo dataset (https://doi.org/10.5281/zenodo.15488704; accessed on 2 July 2025). Taxonomic assignment of the final operational taxonomic units (OTUs) was carried out by comparison of their representative target (partial 5.8S + ITS2; ~150–180 bp for Aspergillus) sequences against the UNITE database v. 9.0 [24] using BLASTn and the lowest common ancestor (LCA) analysis. Aspergillus species identification was refined by additional BLASTn searches at NCBI and double-checking the reference database Mycobank. To evaluate the similarity between Aspergillus species identified by culturing and eDNA, ITS sequences from both isolates and OTUs were pairwise aligned using the BioEdit software.

### 2.6. qPCR on Air Samples

Airborne fungal load was quantified by qPCR as described by García-Gutiérrez et al. [18], using the Sso advanced SYBR Green Supermix (BioRad, Hercules, CA, USA) and the fungi-specific primers NL1f 5′-ATATCAATAAGCGGAGGAAAAG-3′and LS2r 5′-ATTCCCAAACAACTCGACTC-3′ on the 28S rRNA gene [25]. qPCR detection and quantification of *Aspergillus* section *Fumigati* were carried out using the SsoAdvanced Universal Probes Supermix (BioRad) and the specific primers AfumF 5′-CGCGTCCGGTCCTCG-3′ and AfumR 5′-TTAGAAAAATAAAGTTGGGTGTCGG-3′, and the probe AfumP FAM-5′-TGTCACCTGCTCTGTAGGCCCG-3′-TAMRA on the18S rRNA gene [26,27]. qPCR reactions contained 1 × Supermix, 0.4 μM of each primer, and 0.2 μM of probe in a total volume of 10 μL. Standard curves were based on serial dilutions of DNA extracts from reference isolates, an equimolar mixture of *Trichoderma* and *Alternaria* species for the total fungi assay, and the *A. fumigatus* ID633 for the Afum assay. All qPCR trials include positive (two standards in duplicate) and negative reactions, and they were run in a CFX Connect Real-Time PCR Detection System (BioRad) with the following cycling parameters: 95 °C for 5 min followed by 40 cycles consisting of 5 s at 95 °C, 1 min at 60 °C, and subsequent reading of fluorescence at 520 nm. Total fungi assay also includes the final construction of the melting curve. Data analyses were performed using the BioRad CFX Maestro 1.1 software (BioRad). Final results were expressed as target DNA concentration in air (pg/m^3^).

### 2.7. Statistics

Statistical analyses were conducted in R v 4.3.0 through RStudio v 2023.03.0 [28]. Tidyverse v 1.2.1 [29] and vegan v 2.6-4 [30] R packages were used for data manipulation, plotting, and ecological analyses, respectively. The quality-filtered fungal OTU table was rarefied to 13,257 reads per sample using the function *rrarefy* from vegan.

Relative abundance of *Aspergillus* was calculated as percentage of rarefied reads per sample. Considering these data did not follow a normal distribution, based on Shapiro-Wilk test, their variances for different sample types, hospitals, and zones were evaluated by the non-parametric Kruskal-Wallis test and the post-hoc Mann-Whitney-Wilcoxon test. Sequence data from all types of samples (air, vent, HTS, and dust) were analysed together to calculate the relative abundances (percentage of rarefied reads) of the *Aspergillus* species by sampling campaign, hospital, and zone.

## 3. Results

### 3.1. Cultivable Aspergillus Community from Hospitals

Isolation frequencies of the genus *Aspergillus* in the different sampling campaigns, hospitals, and zones are shown in Table 1. This genus was widely distributed in air samples from the three hospitals, covering both outdoor and indoor zones, including some ICUs from LF and SO hospitals. In winter (January–February 2023), similar relative abundances were recorded for outdoor and indoor (IA and IB) zones, mostly in air samples (0.4–21.2%), with a limited representation in surfaces (HTS, vent, and dust). Regardless of the type of sample, the median relative abundance of *Aspergillus* for positive hospital zones in winter was 6.8% of colonies. In autumn (September–October 2023), using a more selective incubation temperature (37 °C instead of 28 °C), we obtained higher relative abundances of airborne *Aspergillus* colonies (14.3–100%; median = 28.3%). Interestingly, for VR and SO hospitals, the proportions of this genus in some indoor zones (admission halls, waiting rooms, and patient rooms) were higher than outdoors, while the LF hospital showed the opposite trend.

A total of 64 isolates of *Aspergillus* (12.5% of the hospital-environment fungal culture collection) have been characterized in this study (Table 2). The majority (55 of 64) were isolated from air samples, while six were isolated from outdoor dust and three from indoor vents. Based on ITS and β-tubulin sequence data, they belong to 23 species of nine sections: *Fumigati* (13 isolates closely related to *A. fumigatus*), *Nidulantes* (13 isolates; *A. sydowii*, *A. unguis*, *A. versicolor*, *A. nidulans*, *A. creber* and *A. stellatus*), *Flavi* (12; *A. flavus*, *A. oryzae* and *A. alliaceus*), *Nigri* (8; *A. niger* and *A. tubingensis*), *Aspergillus* (7; *A. pseudoglaucus*, *A. chevalieri*, *A. montevidensis* and *A. costiformis*), *Usti* (4; *A. calidoustus*, *A. insuetus* and *A. sigurros*), *Circumdati* (3; *A. westerdijkiae*), *Terrei* (3; *A. terreus*), and *Candidi* (1; *A. dobrogensis*). Most *Aspergillus* isolates (60 of 64; 93.7%) have been assigned to species previously reported as pathogens of warm-blooded vertebrates, according to de Hoog et al. [31], including *A. fumigatus*, *A. flavus*, *A. terreus*, *A. niger*, *A. oryzae*, *A. sydowii*, and *A. tubingensis.* All isolates of the *Fumigati* section (13) were identified as *A. fumigatus sensu stricto* and showed azole susceptibility. Susceptibility was tested by an azole–agar screening method [32] using the clinical breakpoints defined by the European Committee on Antimicrobial Susceptibility Testing (EUCAST) [33].

### 3.2. Aspergillus Community Revealed by DNA Metabarcoding

DNA metabarcoding data were achieved for 140 samples, including all air and dust samples, while numerous HTS (68.9%) and vent (43.1%) samples failed in PCR rounds, most likely due to the very low amount of biomass/DNA and the release of PCR inhibitors from the used sponges [18].

Overall, *Aspergillus* was prevalent in the three studied hospitals, detected in 52.8% (74 of 140) of samples, but with limited relative abundances per sample (0.004–38%; median = 1.4%; Figure 1). The highest abundances corresponded to an air sample from an ICU box at VR hospital (VR2_A_IC2; 38%) and a vent sample from an admission hall at LF (LF2_V_IA3; 29.4%), followed by the rest of the samples with relative abundances < 13%. Checking the influence of several factors, we found slight differences in *Aspergillus* abundance depending on the types of sample (higher for HTS), the hospitals (lower for SO), and their zones (higher indoors, increasing with the level of protection: OUT < IA = IB < IC) (Figure 1).

The eDNA-based *Aspergillus* community was composed of 16 OTUs (Figure 2), whose identifications were estimated at the species level considering the closest related sequences from UNITE and NCBI, and similarity between OTU (partial 5.8S + ITS2) and isolate (ITS1 + 5.8S + ITS2) representative sequences. Details of these 16 *Aspergillus* species/OTUs, affiliated to eight sections, (*Fumigati*, *Nidulantes*, *Nigri*, *Circundati*, *Usti*, *Flavi*, *Restricti*, and *Aspergillus*), including their distribution in the study samples and representative ITS2 sequences, are shown in Appendix A. *Aspergillus* species richness was higher in autumn (16) compared to winter (11), as well as in VR (13) and LF (12) hospitals compared to SO (6) (Figure 2, Appendix A). Despite of some specific seasonal differences in the distribution of *Aspergillus* species depending on the hospital, there was no common seasonal pattern for all three hospitals.

The most prevalent and abundant species was *A. pseudoglaucus* (100% similarity to the isolates 250 and 504), which was detected in 45 (32.1%) samples from almost all hospitals, zones, and sampling campaigns, being more dominant in winter than in autumn. This species represented 1.1–100% of the *Aspergillus* community depending on the sampled zone (Figure 2). Only zones lacking this species were OUT at VR and SO, and IC at LF, all three for the autumn campaign. Other species associated with several study zones were: *A. sydowii* (100% similarity to isolates 207 and 236), *A. flavus* (100% similarity to the 8 isolates of this species), *A. versicolor* (100% similarity to isolates 336 and 359), and *A. fumigatus* (100% similarity to 9 of 13 isolates of this species), present in 12, 9, 9, and 4 samples, respectively, from the three hospitals; as well as *A. vitricola*, *A. westerdijkiae* (100% similarity to isolates 194 and 349), *A. halophilicus*, and *A*. *japonicus* detected in 17, 11, 5, and 4 samples, respectively, from VR and LF hospitals.

Some *Aspergillus* species were uniquely detected in one sampling campaign: *A. penicillioides* (2 samples), *A. insuetus* (1; 99.4% similarity to isolates 218 and 576), and *A. venezualensis* (1; 98% similarity to isolate 629) in winter, and *A. fumigatus* (4 samples), *A*. *japonicus* (4), *A. ochraceus* (1), *A. carbonarius* (1), and *A. restrictus* (1) in autumn. In particular, the reference pathogen *A. fumigatus* was very scarce in the eDNA dataset, represented by a few rarefied reads (≤70) in three air samples from regular patient rooms at SO (SO2_A_IB1 and SO2_A_IB3) and an ICU at LF (LF2_A_IC2), and one HTS sample from a patient room at VR (VR2_S_IB3). In addition, *A. tamarii* was detected in two samples from LF, collected in different campaigns (Figure 2, Appendix A).

### 3.3. Quantification of Aspergillus Section Fumigati by qPCR

By using a specific qPCR assay, we only detected seven positive air samples (9.7%; Ct values < 35) for *Aspergillus* section *Fumigati*, mostly collected from outdoors or highly exposed admission halls/waiting rooms, which showed very little target DNA, ranging from 0.7 to 2.7 pg/m^3^. Comparing these data with the total fungal load assessed by qPCR, ratios *Fumigati*/total fungi ranged from 0.002 to 0.05. Five of seven positive samples were also positive for the isolation of *Aspergillus* by culturing [17], with colony relative abundances between 1.5 and 22.8% (Table 3). However, the three samples where *A. fumigatus* was detected by DNA metabarcoding were negative for the *Fumigati*-specific qPCR.

## 4. Discussion

### 4.1. Prevalence and Relative Abundance of Aspergillus in Hospitals

The ubiquitous genus *Aspergillus* is often isolated from many different natural and urban environments, including buildings, being prevalent in indoor air and dust samples [17,34,35,36]. In our study, *Aspergillus* was also widely distributed in the three hospitals, detected in all indoor zones except ICUs from VR hospital, mostly in air samples but also present in some indoor surfaces (vents and HTS). This is likely due to the high sporulation rate in *Aspergillus*, and the efficient aerial dispersion of their conidia (small size, melanised, thermotolerant, hydrophobic, and with slow settling velocity), which remain as airborne for long periods before deposition and after resuspension from the colonized surfaces [2,37].

When it comes to *Aspergillus* relative abundance in the hospital environment, our DNA metabarcoding approach, using fungi specific primers, showed a median abundance of 1.4% of rarefied sequences for the *Aspergillus*-positive samples (52.8%), while the non-selective culture-based approach (on SDCA and DRBC plates incubated at 28 °C) achieved a median of 6.8% of colonies for positive hospital zones. Moreover, when a more selective culture approach (on SDCA at 37 °C) was implemented, the median relative abundance of *Aspergillus* colonies reached 28.3% for positive hospital zones (Table 1). Recognizing the expected higher sensitivity of (PCR-based) eDNA approaches compared to cultivation [18], the opposite trend found here may indicate that *Aspergillus* grows better under standard culture conditions (selected media and incubation temperatures) than other fungi present in the samples. Thus, likely, this feature leads to an overestimation of the proportion of *Aspergillus* in some culture-based environmental studies. Tormo-Molina et al. [38] reported a 4.9% of *Aspergillus* CFU on Sabouraud agar and 6.5% of *Aspergillus-Penicillium* spores directly counted under the light microscope, both on indoor air samples from a Spanish hospital. Lemos et al. [39] counted 7.5% of airborne *Aspergillus* colonies, grown on DRBC plates incubated at 30 °C, from critical zones in a Brazilian hospital. Cabo-Verde et al. [40] reported a 24% of *Aspergillus* isolates after cultivation on malt extract agar at 25 °C of air samples from a hospital in Portugal. Augustowska and Dutkiewicz [41], using a similar approach (Sabouraud agar at 28 °C), reported a clear dominance of *A. fumigatus* in the indoor air mycobiomes from a Polish hospital, representing an average of 77% of isolates. On the other hand, eDNA-based studies also reported different abundances for the airborne *Aspergillus* community inside hospitals. Núñez and García [42] found 24 different *Aspergillus* species with relative abundances ranging from 0.6 to 3.7%, including *A. fumigatus*, *A. flavus*, *A. terreus*, and *A. niger*. Habibi et al. [43] and Chen et al. [44] reported that the *Aspergillaceae* family represented 14.8 and 20.7% of sequences, respectively. According to Tong et al. [45], the *Aspergillus* proportion ranged from 17 to 61% depending on the studied department, with *A. fumigatus* being the most abundant (34–50%) species of this genus.

Interestingly, in this study, the relative abundance of *Aspergillus* sequences was higher indoors and increased with the level of protection of the sampled zone, reaching the highest proportion in ICUs (OUT < IA = IB < IC; Figure 1 right). Previous studies have reported the association of *Aspergillus* with indoor environments (air, dust, and surfaces), overall showing higher abundance there compared to outdoors [34,35,36]. However, to the best of our knowledge, this is the first study reporting a higher proportion of *Aspergillus* in ICUs when compared to other indoor hospital zones. Most likely, this proportional increase is partially due to very low biomass/DNA in the samples collected from these HEPA-protected zones, which leads to a low number of OTUs (fungal richness) from the few DNA copies amplified by PCRs.

### 4.2. Aspergillus Diversity and Its Clinical Relevance

*Aspergillus* genus encloses many species worldwide, many of which are addressed as “cryptic” and grouped in several different sections. The role of these fungi in human diseases is enormously increasing, causing difficult-to-treat infections due to their resistance profiles and demanding a great effort to recognize the involved species [46]. In this study, the *Aspergillus* species found in environmental samples (air and surfaces) from three hospitals in Spain (Sevilla, Valencia, and Madrid) were affiliated with 10 sections: *Fumigati*, *Nidulantes*, *Nigri*, *Terrei*, *Circundati*, *Usti*, *Candidi*, *Flavi*, *Restricti*, and *Aspergillus*, where the latter section accommodated the species formerly described under the genus *Eurotium* [47]. Other environmental studies in hospitals have frequently reported members of the sections *Flavi*, *Fumigati*, *Nidulantes*, *Nigri*, and *Terrei*, although distributed differently, probably depending on the specific sampling sites, climatic determinants, and used methods [7,39,48,49,50].

Summarizing the two sampling campaigns, 64 *Aspergillus* isolates belonging to 23 species of nine sections (excluding the mentioned above *Restricti*) were detected by culturing, while 16 species of eight sections (excluding *Terrei* and *Candidi*) were detected by DNA metabarcoding. Only seven species, *A. fumigatus*, *A. flavus*, *A. sydowii*, *A. pseudoglaucus*, *A. insuetus*, *A. westerdijkiae*, and *A. versicolor*, were detected by both approaches. This relatively low overlap can be due to several possible limiting factors: (i) the already discussed low-moderate relative abundance of *Aspergillus* in the study samples, (ii) difficulty or impossibility to cultivate certain species under the selected media and/or incubation temperatures, (iii) inhibition caused by other cultivated fungi, (iv) observer’s bias during the selection of colony morphotypes, (v) eDNA loss during processing and extraction of samples, and (vi) biases of the selected metabarcoding primers against some species. Based on our previous works characterizing the whole fungal community from these three hospitals [17,18], DNA metabarcoding demonstrated to be much more sensitive than culturing, increasing by about 90–94% the number of observed species (fungal richness). However, when focusing on the genus *Aspergillus*, the opposite trend is observed with a more complete culture-based assessment. This supports the hypothesis mentioned above that environmental relative abundances of *Aspergillus* may be overestimated by culturing due to its relatively good cultivability compared to other fungi.

This study reveals the diversity of potentially pathogenic *Aspergillus* species in three Spanish hospital settings, contributing to the knowledge of the diversity of cryptic species in the hospital environment. Even at low relative abundances, as in this study, the mere presence of this genus in the hospital environment implies a potential health risk, especially in critical zones like ICUs where immunosuppressed patients are receiving care and treatment. In particular, 43.7% (14 of 32) of the *Aspergillus* species identified in this study were detected in ICUs (zone IC). Based on our molecular identifications (closest related sequences from reference databases), many of the reported species have been previously described as causal agents of IPA or other pulmonary disorders: *A. fumigatus*, *A. flavus*, *A. terreus*, *A. niger*, *A. oryzae*, *A. sydowii*, and *A. tubingensis*. Moreover, *A. pseudoglaucus*, *A. alliaceus*, and *A. westerdijkiae* have been described in pulmonary coinfections with other *Aspergillus* species, as well as *A. chevalieri* and *A. unguis*, which may cause otitis and onychomycosis, respectively [31]. In contrast, other species such as *A. sigurros*, *A. insuetus*, *A. dobrogensis*, and *A. vitricola* have not been reported as human pathogens yet.

Despite of the very low relative abundance of *A. fumigatus* in the studied samples, our integrated analysis approach was successful in detecting this species in almost all zones of the three hospitals, including an ICU at LF. The selective culture approach using high incubation temperature (37 °C) demonstrated to be the most effective and sensitive method for the specific detection of this reference pathogen, surpassing the highly sensitive techniques based on eDNA (DNA metabarcoding and qPCR), which showed a few positive samples without consistency between them. Most likely, typical low-biomass/DNA samples from the hospital environment (e.g., air samples from HEPA-filtered rooms and previously cleaned surfaces), together with a low proportion of the target microorganism, lead to the difficult challenge of detecting a few eDNA copies by such techniques. This low environmental abundance of *A. fumigatus* was also reported in other Spanish hospital [42], while a previous 3-year surveillance study, comparing clinical and environmental *Aspergillus* isolates from the SO hospital, reported *A. fumigatus* as the most frequently isolated species (51.9%), followed by *A. niger* (14.9%), *A. terreus* (9.5%), and *A. flavus* (8.4%) [7]. It is worthy to note that the latter study also used a selective temperature for cultivating the environmental samples (Sabouraud agar supplemented with antibacterial compounds at 35 °C). Nevertheless, the simple presence of this hazardous opportunistic pathogen in an ICU at LF hospital deserves further environmental monitoring, as this species was recently included in the critical priority group by WHO [10]. Considering the ever-increasing *Aspergillus* azole resistance in the last decades, and its clinical implications, it is crucial to carry out further surveillance studies on both clinical and environmental isolates. Fortunately, all *A. fumigatus* isolates from this study were susceptible to the tested azoles. Lucio et al. [7] found only two azole-resistant *A. fumigatus* isolates, one clinical and one environmental, in the mentioned 3-year (2019–2021) surveillance study conducted at SO hospital.

According to eDNA profiles, the most prevalent OTU (100% similarity with two isolates) was closely related to *A. pseudoglaucus* and *A. ruber*, both of which have been often isolated from indoor air and dust [51]. *A. ruber* was one of the most abundant airborne fungi in eDNA-based surveys in hospitals from Kuwait and China [43,44]. *A. pseudoglaucus* was reported as the causal agent of sporadic skin infections [52], while *A. ruber* has been associated with IPA cases [53]. Clinical relevance of other prevalent species from this study, detected in several hospitals and zones, is briefly discussed as follows. Two species of the section *Nidulantes*, *A. sidowii* and *A. versicolor*, are frequently detected inside buildings, including hospitals [34,35,39,54], associated with onychomycosis, superficial skin infections, allergies, and production of mycotoxins [31,55,56,57]. *A. versicolor* was also reported as a cause of IPA [58]. Two species of the section *Nigri* (*A. niger* and *A. tubigensis*) and *A. terreus* were uniquely detected by cultivation here. All three are relevant opportunistic pathogens causing IPA, as well as other respiratory and superficial infections [31,39,59,60], and often found in hospital environments [7,39,48,49,50]. *A. flavus* has also been described in the hospital environment [7,39,49,50], with a clear clinical relevance producing mycotoxins and causing allergies and diverse pulmonary diseases, being considered the second-most-important causal agent of IPA [31,61]. *A. westerdijkiae* has been isolated from indoor environments, including sick buildings [35,54,62], and linked to superficial infections and the production of toxic indole alkaloids [52,62]. Finally, *A. vitricola* is a xerophilic species that has been isolated from house dust in Canada and Japan, as well as air and dust samples from museums in Denmark [63,64,65], without known pathogenicity to date.

## 5. Conclusions

*Aspergillus* species are commonly associated with disease in the severely immunocompromised host and those with chronic chest disease. This environmental survey concludes that *Aspergillus* was one of the most prevalent genera in the studied hospitals, though their relative abundance was quite limited. The used integrated approach, combining general and selective culture- and eDNA-based methods, was successful in revealing a considerable diversity of *Aspergillus* (32 species belonging to ten sections) in the hospital environment, and to provide insights about the pros and cons of the different techniques. This combination of eDNA and cultivation methods is essential to overcome the limitations associated with each technique (discussed above) and obtain a more comprehensive picture of the *Aspergillus* community through the integration of complementary datasets. In particular, DNA metabarcoding provides reliable data for characterizing the whole fungal community, including the relative abundances of clinically relevant targets such as *Aspergillus* and *Candida*, while more selective methods (e.g., cultivation at 35–40 °C, or specific qPCR assays for different *Aspergillus* sections) are key to increasing the detection degree of such priority pathogens.

The presence of *Aspergillus* in hospitals is a warning for more stringent control measures to improve their indoor air quality because some of the detected species (*A. fumigatus*, *A. flavus*, and *A. terreus*, among others) are considered regular causal agents of IPA and other diseases in immunocompromised patients. Most of the identified species are potential human pathogens as they have also been isolated from clinical samples in previous studies. As exposure or colonization is a prerequisite to *Aspergillus*-related disease, preventive measures are recommended in healthcare settings. Therefore, it is extremely important to carry out further environmental surveys for monitoring and controlling the fungal burden in hospitals. Detection, isolation, identification, and characterization of fungi from hospital environments are key tasks because most of these fungi have the potential to become pathogenic, particularly in contact with immunocompromised patients. Improvement of available molecular biology methods will allow a widespread use of advanced tools for both microbiological environmental assessment and diagnostics of pathogens.

## Figures and Tables

**Figure 1 jof-11-00626-f001:**
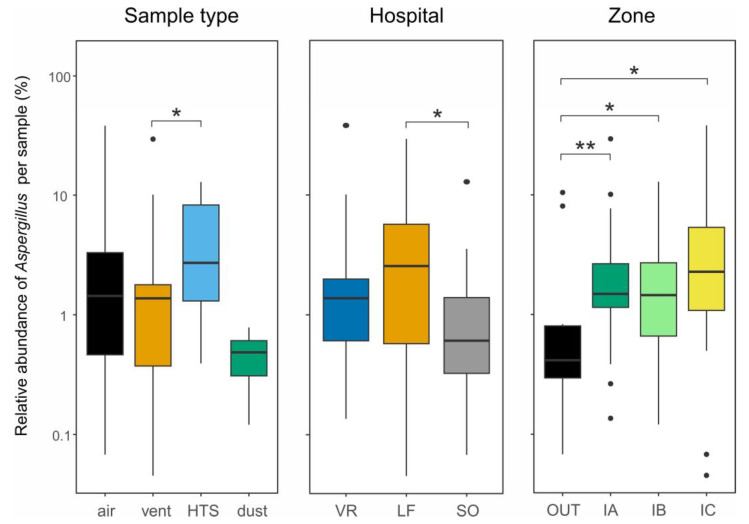
Differences in relative abundance of *Aspergillus* per sample, as percentages of rarefied reads, between sample types (**left**), hospitals (**middle**), and hospital zones (**right**). Asterisks indicate significant differences according to Mann-Whitney-Wilcoxon test (*: *p* ≤ 0.05; **: *p* ≤ 0.01). Lower and upper box boundaries are the 25th and 75th percentiles, respectively; line inside the box is the median; lower and upper lines are whiskers to minimum and maximum values, respectively; and dots are outliers. HTS: high-touch surface; hospitals: VR (Virgen del Rocío), LF (La Fe), and SO (Severo Ochoa); and zones: OUT (outside), IA (unprotected admission halls and waiting rooms), IB (clean regular patient rooms), and IC (intensive care units).

**Figure 2 jof-11-00626-f002:**
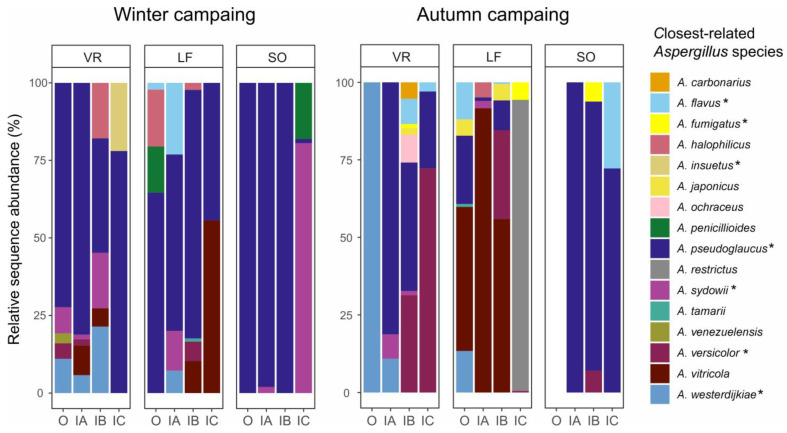
Distribution and relative abundances (percentage of reads) of *Aspergillus* species identified in the different sampling campaigns (winter-left vs. autumn-right), hospitals, and study zones. Sequence data from the different sample types (air, vent, HTS, and dust) were analysed together. Species identifications were assigned based on the closest related sequences from BLASTn searches at NCBI and double-checked using the Mycobank’s taxonomical data. Asterisks indicate species isolated by cultivation (see Table 2). Hospitals: VR (Virgen del Rocío), LF (La Fe), and SO (Severo Ochoa); and study zones: OUT (outside), IA (unprotected admission halls and waiting rooms), IB (clean regular patient rooms), and IC (intensive care units).

**Table 1 jof-11-00626-t001:** Relative abundance (%) of *Aspergillus* colonies grown from environmental samples from different hospitals, zones, and sampling campaigns. Data extracted from [17].

Sample Types ^1^	VR ^2^					LF ^2^					SO ^2^			
	OUT	IA	IB	IC		OUT	IA	IB	IC		OUT	IA	IB	IC
Winter														
Air	18.8	12.2	21.2	0		13.1	9.5	12.5	8.3		0.4	3.7	5.3	2.4
HTS	-	4.5	0	0		-	0	0	0		-	0	0	0
Vent	-	17.9	24.3	0		-	0.2	0	0		-	4.5	0	0
Dust	1.6	-	-	-		1.3	-	-	-		0	-	-	-
Autumn														
Air	25	100	64.3	0		60	14.3	25	25		19.3	31.6	75.7	0

^1^ Winter samples (air, HTS, vent, and dust) were incubated on SDCA and DRBC at 28 °C, while autumn samples (only air) were incubated on SDCA at 37 °C. HTS: high-touch surfaces. ^2^ Origin of samples: VR (Virgen del Rocío), LF (La Fe), and SO (Severo Ochoa) hospitals; OUT (outside), IA (unprotected admission halls and waiting rooms), IB (clean regular patient rooms), and IC (intensive care units).

**Table 2 jof-11-00626-t002:** *Aspergillus* isolates characterized in this study.

ID ^1^	Camp. ^2^	Hosp. ^3^	Zone ^4^	Sample	ITS Closest Related ^5^	Ident. (%)	ITS Acc. No.	*β*-Tubulin Closest Related ^5^	Ident. (%)	*β*-Tubulin Acc. No.	Infect. ^6^
Section *Aspergillus*											
250	W	VR	OUT	Air	*A. pseudoglaucus*	100	PP844117	*A. pseudoglaucus*	100	PV345508	+
252	W	VR	OUT	Air	*A. ruber*	96.8	PP844119	*A. chevalieri*	100	PV345509	+
270	W	VR	OUT	Air	*A. chevalieri*	100	PP844135	*A. chevalieri*	100	PV345511	+
382	W	LF	OUT	Air	*A. montevidensis*	99.1	PP844236	*A. montevidensis*	100	PV345523	+
404	W	LF	IA	Vent	*A. montevidensis*	100	PP844256	*A. montevidensis*	100	PV345528	+
504	W	SO	IA	Air	*A. pseudoglaucus*	100	PP844355	*A. pseudoglaucus*	100	PV345532	+
579	W	SO	IC	Air	*A. costiformis*	99.8	PP844430	*A. costiformis*	100	PV345538	+
Section *Candidi*											
586	W	SO	IC	Air	*A. candidus*	100	PP844437	*A. dobrogensis*	99.81	PV345540	-
Section *Circumdati*											
194	W	VR	IA	Air	*A. westerdijkiae*	100	PP844062	*A. westerdijkiae*	100	PV345500	+
292	W	VR	IA	Dust	*A. westerdijkiae*	99.6	PP844157	*A. westerdijkiae*	99.8	PV345514	+
349	W	VR	IB	Air	*A. westerdijkiae*	100	PP844210	*A. westerdijkiae*	100	PV345519	+
Section *Flavi*											
191	W	VR	IA	Air	*A. flavus*	99.6	PP844059	*A. flavus*	100	PV345498	+
192	W	VR	IA	Air	*A. flavus*	100	PP844060	*A. flavus*	100	PV345499	+
202	W	VR	IB	Air	*A. oryzae*	99.6	PP844070	*A. oryzae*	100	PV345502	+
267	W	VR	OUT	Air	*A. flavus*	100	PP844132	*A. flavus*	100	PV345510	+
608	A	VR	IB	Air	*A. flavus*	100	PP844459	nd ^7^			+
375	W	LF	OUT	Dust	*A. flavus*	100	PP844231	*A. flavus*	100	PV345522	+
390	W	LF	OUT	Air	*A. flavus*	100	PP844243	*A. flavus*	100	PV345526	+
631	A	LF	OUT	Air	*A. flavus*	99.8	PP844473	*A. flavus*	100	PV345547	+
632	A	LF	OUT	Air	*A. flavus*	99.8	PP844474	*A. flavus*	100	PV345548	+
657	A	SO	OUT	Air	*A. oryzae*	100	PP844498	*A. oryzae*	100	PV345552	+
679	A	SO	IB	Air	*A. flavus*	94.4	nd ^8^	*A. flavus*	100	PV345559	+
680	A	SO	IB	Air	*A. alliaceus*	100	PP844518	*A. alliaceus*	99.6	PV345560	+
Section *Fumigati*											
195	W	VR	IA	Air	*A. fumigatus*	99.8	PP844063	*A. fumigatus*	100	PV345501	+
341	W	VR	IB	Air	*A. fumigatus*	100	PP844202	*A. fumigatus*	100	PV345518	+
630	A	LF	OUT	Air	*A. fumigatus*	100	PP844472	*A. fumigatus*	100	PV345546	+
633	A	LF	IA	Air	*A. fumigatus*	100	PP844475	*A. fumigatus*	100	PV345549	+
637	A	LF	IB	Air	*A. fumigatus*	100	PP844479	*A. fumigatus*	100	PV345550	+
641	A	LF	IC	Air	*A. fumigatus*	100	PP844483	*A. fumigatus*	100	PV345551	+
498	W	SO	IA	Air	*A. fumigatus*	99.6	PP844349	*A. fumigatus*	100	PV345531	+
536	W	SO	IB	Air	*A. fumigatus*	100	PP844387	*A. fumigatus*	100	PV345534	+
549	W	SO	IB	Air	*A. fumigatus*	100	PP844400	*A. fumigatus*	100	PV345535	+
561	W	SO	IB	Air	*A. fumigatus*	99.8	PP844412	*A. fumigatus*	100	PV345536	+
671	A	SO	IA	Air	*A. fumigatus*	100	PP844510	*A. fumigatus*	100	PV345555	+
675	A	SO	IB	Air	*A. fumigatus*	99.3	PP844514	*A. fumigatus*	100	PV345556	+
678	A	SO	IB	Air	*A. fumigatus*	99.6	PP844517	*A. fumigatus*	100	PV345558	+
Section *Nidulantes*											
207	W	VR	IB	Air	*A. sydowii*	100	PP844075	*A. sydowii*	100	PV345503	+
236	W	VR	IC	Air	*A. sydowii*	100	PP844103	*A. sydowii*	100	PV345506	+
276	W	VR	OUT	Dust	*A. unguis*	99.6	PP844141	*A. unguis*	98.9	PV345512	+
328	W	VR	IB	Air	*A. sydowii*	99.6	PP844189	*A. sydowii*	100	PV345516	+
336	W	VR	IB	Vent	*A. versicolor*	99.6	PP844197	*A. versicolor*	100	PV345517	+
355	W	VR	IB	Air	*A. sydowii*	99.8	PP844216	*A. sydowii*	100	PV345520	+
359	W	VR	IB	Air	*A. versicolor*	100	PP844220	*A. versicolor*	100	PV345521	+
387	W	LF	OUT	Dust	*A. nidulans*	100	PP844240	*A. nidulans*	99.2	PV345525	+
394	W	LF	OUT	Dust	*A. unguis*	100	PP844247	*A. unguis*	98.6	PV345527	+
450	W	LF	IC	Air	*A. creber*	100	PP844301	*A. creber*	100	PV345529	+
629	A	LF	OUT	Air	*A. stellatus*	100	PP844471	*A. stellatus*	100	PV345545	+
531	W	SO	IA	Vent	*A. sydowii*	98.8	PP844382	*A. sydowii*	100	PV345533	+
587	W	SO	IC	Air	*A. sydowii*	100	PP844438	*A. sydowii*	100	PV345541	+
Section *Nigri*											
211	W	VR	IB	Dust	*A. niger*	100	PP844079	*A. niger*	100	PV345504	+
238	W	VR	IC	Air	*A. tubingensis*	98.6	PP844105	*A. tubingensis*	99.8	PV345507	+
325	W	VR	IB	Air	*A. tubingensis*	100	PP844186	*A. tubingensis*	98.6	PV345515	+
605	A	VR	IA	Air	*A. niger*	99.8	PP844456	*A. tubingensis*	100	PV345542	+
606	A	VR	IB	Air	*A. tubingensis*	100	PP844457	*A. tubingensis*	100	PV345543	+
384	W	LF	OUT	Air	*A. niger*	100	PP844238	*A. niger*	100	PV345524	+
468	W	SO	OUT	Air	*A. niger*	100	PP844319	*A. niger*	100	PV345530	+
664	A	SO	IA	Air	*A. niger*	99.8	PP844504	*A. niger*	100	PV345553	+
Section *Terrei*											
283	W	VR	IA	Air	*A. terreus*	100	PP844148	*A. terreus*	100	PV345513	+
628	A	LF	OUT	Air	*A. terreus*	100	PP844470	*A. terreus*	99.8	PV345544	+
669	A	SO	IA	Air	*A. terreus*	92.1	nd ^8^	*A. terreus*	100	PV345554	+
Section *Usti*											
218	W	VR	ID	Air	*A. calidoustus*	100	PP844085	*A. calidoustus*	100	PV345505	+
576	W	SO	IC	Air	*A. insuetus*	99.6	PP844427	*A. insuetus*	100	PV345537	-
580	W	SO	IC	Air	*A. sigurros*	99.8	PP844431	*A. sigurros*	100	PV345539	-
676 ^9^	A	SO	IB	Air	*A. flavus*	99.8	PP844515	*A. insuetus*	100	PV345557	-

^1^ Isolates are sorted by *Aspergillus* sections, which were determined based on both ITS and β-tubulin data, with the latter marker being more informative. ^2^ Sampling campaigns: W (winter) and A (autumn). Winter samples were incubated on SDCA and DRBC at 28 °C, while autumn samples (only air) were incubated on SDCA at 37 °C. ^3^ Study hospitals: VR (Virgen del Rocío), LF (La Fe), and SO (Severo Ochoa). ^4^ Study zones: OUT (outside), IA (unprotected admission halls and waiting rooms), IB (clean regular patient rooms), and IC (intensive care units). ^5^ The species identifications were assigned based on the closest related sequences from BLASTn searches at NCBI and double-checked using Mycobank’s taxonomical data. ^6^ Positive or negative reported infections in warm-blooded vertebrates according to [31]. ^7^ Not determined (nd) because of the failed PCR. ^8^ Not determined (nd) because of the low quality of sequences. ^9^ Uncertain taxonomical assignment of the isolate ID 676 due to the lack of agreement between ITS and β-tubulin data. This is shown in the section *Usti* based on β-tubulin result.

**Table 3 jof-11-00626-t003:** qPCR results for air samples with positive detection of *Aspergillus* section *Fumigati*.

Samples	Hospital ^1^	Zone ^2^	Section *Fumigati* DNA (pg/m^3^)	Total Fungal DNA (pg/m^3^)	Ratio *Fumigati*/Total Fungi	Relative Abundance of *Aspergillus* Colonies ^3^ (%) [17]	
						SDCA	DRBC
VR_A_O3	VR	OUT	1	39.3	0.02	17.5	3
VR_A_IA1	VR	IA	0.8	95.6	0.01	20.5	22.8
SO_A_IA1	SO	IA	2.7	55	0.05	18.2	20
SO_A_IA2	SO	IA	1.5	172.8	0.01	0	0
SO_A_IA3	SO	IA	0.7	310.4	0.002	0	0
SO_A_IB1	SO	IB	1.1	71.9	0.01	1.5	4.3
LF2_A_O1	LF2	OUT	0.7	16	0.04	nd	nd

^1^ Study hospitals: VR (Virgen del Rocío), LF (La Fe), and SO (Severo Ochoa). ^2^ Study zones: OUT (outside), IA (unprotected admission halls and waiting rooms), IB (clean regular patient rooms), and IC (intensive care units). ^3^ Relative abundances of *Aspergillus* colonies cultivated on SDAC and DRBC media at 28 °C, only determined for winter samples. nd: not determined.

## Data Availability

Raw sequencing data are available at the European Nucleotide Archive (ENA), EMBL-EBI, under the study accession no. PRJEB86993 (https://www.ebi.ac.uk/ena/browser/view/PRJEB86993; accessed on 2 July 2025). More details about the bioinformatics and statistical analyses of the complete DNA metabarcoding dataset are available at Zenodo (https://doi.org/10.5281/zenodo.15488704; accessed on 2 July 2025).

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
