# Peer review of "An Integrated Analysis Approach to Unravel the Aspergillus Community in the Hospital Environment"

_jof, 2025, doi:10.3390/jof11090626_

Round 1

Reviewer 1 Report

The manuscript should justify better why is important to use the molecular tools applied in light of the results when comparing with culture results.

Discuss the limitations of molecular methods in detail, particularly in cases where cultures yielded Aspergillus but molecular testing did not detect it. 

Justify the sentence in line 435: "This supports that..."

Author Response

Comments 1:

Does the title describe the article's topic with sufficient precision?

No

I would suggest to include "An integrated analyses approach..." since the paper focus on combining different assays

Response 1: We have edited the title following the reviewer’s suggestion. Now the article title is “An integrated analysis approach to unravel the Aspergillus community in the hospital environment”. In addition, we have changed “integrated approach” by “integrated analysis approach” in line 457.

Comments 2:

Does the introduction provide a comprehensive yet concise overview about the state of knowledge in the area of research?

No

Some information is needed regarding the reason of incubation at 37ºC

Response 2: We have slightly edited the corresponding sentence in Material and Methods to clarify this point (lines 115-117): “In autumn, to selectively cultivate potentially pathogenic Aspergillus species, air samples were plated on SDCA with and without 4mg/L voriconazole and incubated at 37°C.”

However, we do not see the need of adding in Introduction any information about this methodological detail.

Comments 3:

Major comments

The manuscript should justify better why is important to use the molecular tools applied in light of the results when comparing with culture results.

Response 3: We agree that this point should be better justified. Therefore, to clarify it, we have added the following statement in Conclusions (lines 510-518):

“The used integrated approach, combining general and selective culture- and eDNA-based methods, was successful to reveal a considerable diversity of Aspergillus (32 species belonging to ten sections) in the hospital environment, and to provide insights about pros and cons of the different techniques. This combination of eDNA and cultivation methods is essential to overcome the limitations associated with each technique (discussed above), and to get a more comprehensive picture of the Aspergillus community through the integration of complementary datasets. In particular, DNA metabarcoding provides reliable data for characterizing the whole fungal community, including the relative abundances of clinically relevant targets such Aspergillus and Candida; while more selective methods (e.g., cultivation at 35 – 40 °C, or specific qPCR assays for different Aspergillus sections) are key to increase detection degree of such priority pathogens.”

Comments 4:

Detailed comments

Discuss the limitations of molecular methods in detail, particularly in cases where cultures yielded Aspergillus but molecular testing did not detect it.

Response 4: See the response to the comments 3 above. The limitations of eDNA-based methods are already discussed in several parts of the Discussion:

Lines 428 – 434: “This relatively low overlap can be due to several possible limiting factors: …. (v) eDNA loss during processing and extraction of samples, and (vi) biases of the selected metabarcoding primers against some species.”

Lines 458 - 465: “The selective culture approach using high incubation temperature (37 °C) demonstrated to be the most effective and sensitive method to the specific detection of this reference pathogen, surpassing the highly sensitive techniques based on eDNA (DNA metabarcoding and qPCR), which showed a few positive samples without consistency between them. Most likely, typical low biomass/DNA samples from the hospital environment (e.g. air samples from HEPA-filtered rooms and previously cleaned surfaces) together a low proportion of the target microorganism lead to the difficult challenge of detecting a few eDNA copies by such techniques.”

Comments 5:

Justify the sentence in line 435: "This supports that..."

Response 5: In our opinion, the mentioned sentence is well justified considering the preceding statements (detailed below). We have added the words “mentioned above” to clarify the link between the two statements:

Lines 380 - 384 (Section 4.1): “Recognizing the expected higher sensitivity of (PCR-based) eDNA approaches compared to cultivation [18], the opposite trend found here may indicate that Aspergillus grows better under standard culture conditions (selected media and incubation temperatures) than other fungi present in the samples. Thus, likely, this feature leads to an overestimation of the proportion of Aspergillus in some culture-based environmental studies.”

Lines 434 - 441 (Section 4.2, including the commented sentence): “Based on our previous works characterizing the whole fungal community from these three hospitals [17,18], DNA metabarcoding demonstrated to be much more sensitive than culturing, increasing about 90 – 94 % the number of observed species (fungal richness). However, when focusing on the genus Aspergillus, the opposite trend is observed with a more complete culture-based assessment. This supports the hypothesis mentioned above that environmental relative abundances of Aspergillus may be overestimated by culturing due to its relatively good cultivability compared to other fungi.”

Reviewer 2 Report

Overall, the article “An integrated approach to unravel the Aspergillus community in the hospital environmentis“ is very interesting and important in terms of surveillance of potentially pathogenic fungi; it is written in a good language, well structured and illustrated and contains valuable comparative data on the use of different methods to detect Aspergillus.

Minor points that should be clarified/corrected:

  1. Why was it decided to take samples from the vent, but not from the HEPA filters.
  2. Lines 107-108 “18 settled dust from outdoor surfaces” Please specify from which surfaces the samples were taken.
  3. Please add sentences in the Discussion section regarding sensitivity to the azoles tested (although all A. fumigatus isolates from this study were sensitive to the azoles tested, the data should be compared to the results of other studies); add a few sentences to discuss seasonal differences. These data are listed and illustrated in the results but are not discussed.

      4 Please remove the reference in the Conclusion.

Overall, the article “An integrated approach to unravel the Aspergillus community in the hospital environmentis“ is very interesting and important in terms of surveillance of potentially pathogenic fungi; it is written in a good language, well structured and illustrated and contains valuable comparative data on the use of different methods to detect Aspergillus.

Minor points that should be clarified/corrected:

  1. Why was it decided to take samples from the vent, but not from the HEPA filters.
  2. Lines 107-108 “18 settled dust from outdoor surfaces” Please specify from which surfaces the samples were taken.
  3. Please add sentences in the Discussion section regarding sensitivity to the azoles tested (although all A. fumigatus isolates from this study were sensitive to the azoles tested, the data should be compared to the results of other studies); add a few sentences to discuss seasonal differences. These data are listed and illustrated in the results but are not discussed.

      4 Please remove the reference in the Conclusion.

Author Response

Comments 1:

Major comments

Overall, the article “An integrated approach to unravel the Aspergillus community in the hospital environment” is very interesting and important in terms of surveillance of potentially pathogenic fungi; it is written in a good language, well structured and illustrated and contains valuable comparative data on the use of different methods to detect Aspergillus.

Response 1: Many thanks for your positive comments.

Comments 2:

Minor points that should be clarified/corrected:

Why was it decided to take samples from the vent, but not from the HEPA filters.

Response 2: Unfortunately, this study could not cover the filters of HVAC (Heating, ventilation and Air Conditioning) systems, which also are relevant in microbiological environmental assessments. Here we studied/compared air and surface samples from different zones. For indoor zones, we selected two contrasting types of surface samples, high-touch surfaces and vents, the latter samples include the particles deposited on the intake ventilation grids. They are representative of the bioaerosols transported by the intake air, and are highly relevant for patient exposure.

Comments 3:

Lines 107-108 “18 settled dust from outdoor surfaces” Please specify from which surfaces the samples were taken.

Response 3: Now this is detailed in the corresponding sentence (lines 107-108): “…, and 18 settled dust from outdoor surfaces such as window sills and railings.”

Comments 4:

Please add sentences in the Discussion section regarding sensitivity to the azoles tested (although all A. fumigatus isolates from this study were sensitive to the azoles tested, the data should be compared to the results of other studies); add a few sentences to discuss seasonal differences. These data are listed and illustrated in the results but are not discussed.

Response 4: The global problem of the Aspergillus azole resistance is already highlighted in two sections:

Introduction (lines 55 - 63): “Triazole drugs are the antifungal of choice for prophylaxis and first-line treatment of Aspergillus infections. However, azole-resistance reports in both clinical and environmental isolates of A. fumigatus have continuously increased in the last decades [7]. The omnipresent use of azoles in agriculture, horticulture and forest industry is thought to be the main cause of the emergence of azole cross-resistant A. fumigatus isolates from the environment [8,9]. This critical situation has motivated the inclusion of this species in the critical priority group of the first fungal priority pathogen list developed by the World Health Organization (WHO) in 2022 [10].”

Discussion (474 - 477): “Considering the ever-increasing Aspergillus azole resistance in the last decades, and its clinical implications, it is crucial to carry out further surveillance studies on both clinical and environmental isolates. Fortunately, all A. fumigatus isolates from this study were susceptible to the tested azoles.

Following the reviewer’s suggestion, we have added this sentence to the discussion about the previous study carried out at SO hospital (lines 478 - 480):

Lucio et al. [7] only found two azole-resistant A. fumigatus isolates, one clinical and one environmental, in the mentioned 3-year (2019 – 2021) surveillance study conducted at SO hospital.”

As already noted by the reviewer, all A. fumigatus isolates from this study were azole susceptible. Moreover, given the limited number of isolates (13) it is not surprising that we did not found any azole resistant strain. Therefore, we do not see any need of discussing further this point comparing to other specific studies.

When it comes to the seasonal differences, we did not observe any clear pattern between the two sampling campaigns, except the slightly higher Aspergillus richness in autumn, and higher dominance of A. pseudoglaucus in winter. We clarified this point adding / editing the following statements in Results (lines 309-314): “Despite of some specific seasonal differences in the distribution of Aspergillus species depending on the hospital, there was no common seasonal pattern for all three hospitals.

The most prevalent and abundant species was A. pseudoglaucus (100% similarity to the isolates 250 and 504), which was detected in 45 (32.1 %) samples from almost all hospitals, zones and sampling campaigns, being more dominant in winter than in au-tumn.”

Comments 5:

Please remove the reference in the Conclusion.

Response 5: The reference [66] has been removed from both Conclusions and  reference list.

Round 2

Reviewer 1 Report

The authors complied with the comments and the MS should be accepted. 

The authors complied with the comments and the MS should be accepted.